# SGAT: Shuffle and graph attention based Siamese networks for visual tracking

**Jun Wang**[1,2], **Limin Zhang**[1,2], **Wenshuang Zhang**[1,2], **Yuanyun Wang**[1,2]*, **Chengzhi Deng**[1,2]

1 School of Information Engineering, Nanchang Institute of Technology, Nanchang, Jiangxi, China, 2 Jiangxi Province Key Laboratory of Water Information Cooperative Sensing and Intelligent Processing, Nanchang Institute of Technology, Nanchang, Jiangxi, China

* wangyy_abc@163.com

**Data Availability Statement:** (1) OTB2015: Wu Y, Lim J, Yang MH. Object Tracking Benchmark. In: IEEE Transactions on Pattern Analysis and Machine Intelligence; 2015. p. 1834–1848. DOI: 10.1109/TPAMI.2014.2388226 URL: https://ieeexplore.ieee.

## Abstract

Siamese-based trackers have achieved excellent performance and attracted extensive attention, which regard the tracking task as a similarity learning between the target template and search regions. However, most Siamese-based trackers do not effectively exploit correlations of the spatial and channel-wise information to represent targets. Meanwhile, the cross-correlation is a linear matching method and neglects the structured and part-level information. In this paper, we propose a novel tracking algorithm for feature extraction of target templates and search region images. Based on convolutional neural networks and shuffle attention, the tracking algorithm computes the similarity between the template and a search region through a graph attention matching. The proposed tracking algorithm exploits the correlations between the spatial and channel-wise information to highlight the target region. Moreover, the graph matching can greatly alleviate the influences of appearance variations such as partial occlusions. Extensive experiments demonstrate that the proposed tracking algorithm achieves excellent tracking results on multiple challenging benchmarks. Compared with other state-of-the-art methods, the proposed tracking algorithm achieves excellent tracking performance.

## 1 Introduction

Visual tracking [1–3] is a fundamental research topic in computer vision. It aims to estimate target states in subsequent frames by given the initial state in the first frame. It is widely used in various applications, such as video surveillance [4], human-computer interaction [5], augmented reality [6], and so on. Recently, Convolutional Neural Network (CNN) is successfully used in visual tracking. Deep trackers [7, 8] achieve robust tracking performance and real-time tracking speed. However, due to complicated appearance variations, visual tracking is still a challenging task.

In recent years, Siamese network is widely used in visual tracking. Siamese-based trackers regard tracking problems as a similarity measuring between the template patch and a search patch. SiamFC [1] applies Siamese network to visual tracking for the first time, and achieves

org/document/7001050 (2) GOT-10k: Huang L, Zhao X, Huang K. Got-10k: A large high-diversity benchmark for generic object tracking in the wild. In: IEEE Transactions on Pattern Analysis and Machine Intelligence; 2019. p. 1562–1577. DOI: 10.1109/TPAMI.2019.2957464 URL: https://ieeexplore.ieee.org/abstract/document/8922619 (3) LaSOT: Fan, H, Ling, H, Lin, L, Yang, F. LaSOT: A High-Quality Benchmark for Large-Scale Single Object Tracking. In: IEEE Conference on Computer Vision and Pattern Recognition. 2019, p. 5374–5383. DOI: 10.1109/cvpr.2019.00552 URL: https://doi.org/10.48550/arXiv.1809.07845 (4) UAV123: M. Mueller, N. Smith, B. Ghanem. A benchmark and simulator for uav tracking. In: European conference on computer vision, Springer, 2016, p. 445–461. DOI: 10.1007/978-3-319-46448-0_27 URL: http://dx.doi.org/10.1007/978-3-319-46448-0_27.

**Funding:** Yuanyun Wang, Wenshuang Zhang, Limin Zhang are funded by the Jiangxi Science and Technology Research Project of Education within the Department of China (No: GJJ190955), and the National Natural Science Foundation of China (No: 61861032) for the study design, the experiments and the paper publishing. Jun Wang is funded by the National Natural Science Foundation of China (No: 61865012) for the study and the publication.

**Competing interests:** The authors have declared that no competing interests exist.

excellent tracking performance. Especially, it has a good balance between tracking accuracy and real-time tracking speed. Based on SiamFC, DSiam [9] learns target appearance variations to improve adaptation capabilities. SiamPCF [10] proposes a novel anchor-free visual tracking algorithm, which use points instead of bounding box to describe a target and achieves excellent tracking performance. FSNet [11] develops a feature selection convolutional neural network to reduce the computational complexity and accelerate offline training.

Inspired by region proposal network (RPN) [12], Li *et al.* [13] considers visual tracking as two subtasks including the classification and regression. Later works [14, 15], improve SiamRPN in tracking accuracy and reduce redundant parameters. These RPN-based trackers use anchors to obtain candidate boxes. However, these trackers are sensitive to the size, number and aspect ratios of anchor boxes. Also, the tuning of hyper-parameters is time-consuming. To solve the above problems, recently, anchor-free based trackers such as SiamFC++ [16] and SiamCAR [17] are proposed. All these anchors-free trackers discard anchors and proposals, and greatly reduces the time-consuming on tuning of hyper-parameters.

It is worth mentioning that both anchor and anchor-free based trackers use a powerful CNN such as AlexNet [18], ResNet [19] and GoogLeNet [20] to extract features in a Siamese network. However, most Siamese-based trackers use the features of the last convolution layer or cascaded multi-layers as the target representations of the template and the search region, which do not effectively use the structured and part-level information. Motivated by these considerations, attention mechanism is applied in CNN to improve the feature representation, which focuses on essential appearance features while distractors.

Another core component of Siamese-based trackers is the similarity learning. SiamFC introduces the Siamese network as the feature extractor and adopts the cross-correlation operator to compute the similarity between the target template and a search region. In SiamPRN++, a depth-wise correlation is used to reduce the number of model parameters and makes the off-line training model more stable. However, both the cross-correlation and depth cross-correlation take the template features as a whole for linear matching on the search regions, so that the adjacent sliding windows produce a similar response. Guo *et al.* [21] propose a graph attention module(GAM) to realize a part-to-part matching between the template and a search region.

Different from the previous work [22], we design a novel feature extraction network based on GoogleNet to exploit correlations of the spatial and channel-wise information. Additionally, in order to alleviate the influences of appearance variations, we use a different similarity computing to obtain more accurate score maps. Inspired by above-mentioned works, in this paper, we propose a novel tracking algorithm based on shuffle attention mechanism and graph matching in Siamese network. The shuffle attention mechanism in the backbone network reconstructs the basic features extracted from CNN, and makes the feature representation focusing on the regions of interest through spatial and channel-wise transformations. Different from the cross-correlation based similarity learning, the part-to-part graph attention matching further improves the tracking robustness in complex scenes, such as occlusion.

The contribution of this paper can be summarized as follows:

- We propose an end-to-end deep model based on CNN and shuffle attention unit to enhance the capacity of feature representations. The model effectively exploits the correlations of the spatial and channel-wise information without extra overhead.

- We develop a novel tracking framework based on Siamese network, consisting of the designed deep model, graph attention matching and prediction head. Compared with the traditional cross-correlation based trackers, the proposed tracking algorithm exploits

structured and part-level information, which greatly alleviate the influences of appearance variations such as fast motion and partial occlusions.

- Extensive experimental results demonstrate that the proposed tracker has excellent performance on multiple benchmarks including OTB-100 [23], GOT-10k [24], UAV123 [25] and LaSOT [26], and outperforms many SOTA trackers. At the same time, the proposed tracking algorithm meets the real-time requirement with an average speed of 60 FPS.

The rest of this paper is arranged as follows. In Section 2, we review the related works. The details of the proposed method are described in Section 3. The experimental results on four benchmarks are presented followed by the ablation study and qualitative evaluations in Section 4. At last, we draw a conclusion in Section 5.

## 2 Related works

In this section, we mainly review some representative works and techniques that are closely related to the proposed algorithm including Siamese-based tracking algorithms and some attention mechanisms.

### 2.1 Siamese-based visual tracking

In recent years, trackers based on Siamese network attract incremental attention for their leading performance [27–29]. Tao *et al*. [30] propose to learn a similarity function to locate a target. However, because the processing of candidate sampling is not efficient enough, SINT [30] cannot meet the real-time tracking speed (about 2FPS). In [1], a fully convolution Siamese network framework is proposed to compute the similarity between the target template and a search region in an embedding space, which achieves beyond real-time tracking. Many later works regard SiamFC as guideline to improve tracking accuracy. Guo *et al*. [9] propose a dynamic Siamese network that can effectively learn the appearance variations of a target and suppress the background information through a fast transformation learning model.

Fan *et al*. [31] propose a dual-margin model for accuracy and robust visual tracking, which formulated the target state prediction problem as a dual-margin model including an intra-object margin and an inter-object margin. Li *et al*. [32] propose a thermal infrared tracker based on a hierarchical spatially-aware twin network that regards the infrared tracking problem as a similarity verification task.

Recently, Li *et al*. [13] propose to connect the region proposal extraction subnetwork to the Siamese network framework, which distinguishes the target from the surrounding background by classification branch and estimate the bounding box by regression branch. SiamRPN [13] achieves promising tracking accuracy and beyond real-time speed. Many outstanding works such as DaSiamRPN [33], and C-RPN [34] use SiamRPN as a baseline, and improve the tracking accuracy by designing different network models.

However, most of the Siamese network trackers connecting RPN subnetworks are based on anchors, which are sensitive to the size, number and aspect ratios of anchor boxes. Xu *et al*. [16] analyze the prior tracking methods and propose a set of practical target state estimation criteria. It effectively solves wrong matching problem due to anchor objects. Guo *et al*. [17] propose a new fully convolution Siamese network framework to solve the end-to-end tracking problem in a per-pixel manner, which framework is both proposal and anchor-free. Zhang *et al*. [35] propose a novel object-aware and anchor-free networks. Compared with anchor-based trackers, it can regress to the target region in a large spatial range. Under a fully convolution network, Chen *et al*. [36] propose Siamese Box Adaptive Network for object tracking, and regard the tracking task as a parallel classification and regression problem.

## 2.2 Attention mechanism

Recent works [37, 38] have made many contributions in improving the performance of convolutional neural networks, and these efforts can be roughly divided into two categories. The first category aims to improve the performance of CNN from the spatial domain. The representative works mainly include dilated convolution [39, 40] and deformation convolution [41, 42]. The above works mainly expands the receptive field of the networks by using the predefined gap, and mainly focuses on the target region adaptively through a series of row and column transformations. The disadvantages of these methods are that they do not make use of the clue information between channels.

The second category joints space transformation and channel attention to enhance the performance of CNN. Representative works include SENet [38], CBAM [43], and so on. These methods usually use additional subnetworks to redistribute the weight of the feature map. These networks highlight the target region and suppress the background interference, and achieve excellent feature representation performance. However, these methods usually have additional parameters and overhead.

By jointing feature extraction network and attention mechanism to represent the template and search region patch, Siamese-based algorithms have achieved significant performance improvement. Zhang *et al*. [19] propose a residual attentional Siamese network (RASNet) to adaptively reconstruct a model. He *et al*. [2] build a twofold Siamese network (SA-Siam) to represent the appearance and semantic information, and channel attention is introduced to enhance the capture of semantic information. Hua *et al*. [44] propose a lightweight UAV algorithm based on attention mechanism and strategy gradient to improve overall tracking accuracy and robustness. Different from these works, we reconstruct the basic features by combining spatial and channel attentions. Finally, a nonlinear graph attention matching is used instead of cross-correlation to capture spatial semantic information.

## 3 Method

In this section, we firstly introduce the SGAT algorithm in detail. As shown in Fig 1, it includes three main components: 1) feature extraction network with shared weight, which is used for depth feature extraction of the target template and search regions; 2) the shuffle attention mechanism model (SA Unit), which reconstructs the basis features to focus on the target region and suppress the background interference through the spatial and channel-wise transformation; 3) graph attention matching (GM), which computes the similarity between the target template and a search region, and joints classification and regression branches to locate the target position in the current frame.

## 3.1 Siamese-based object tracking

The trackers based on Siamese network have become more popular due to their leading tracking performance [17, 45]. These trackers regard the tracking task as a similarity matching problem between the target template and a search region, and greatly improves the accuracy and robustness.

In Siamese-based trackers, firstly, the initial frame is preprocessed to obtain the template image *x* and the search images *z* are cropped in the subsequent frames. Then, these image patches are input into the convolution neural network with shared weights for feature extraction. Nextly, the similarity between the template feature and search region features are measured through cross-correlation, and the score maps are obtained. Finally, the target center position offset and scale change are calculated according to the value of the score map. The

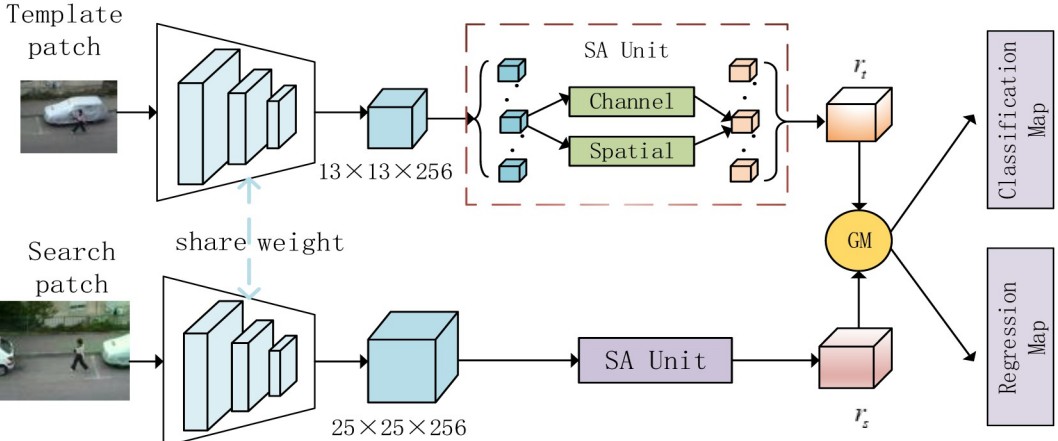

**Fig 1. Architecture of the proposed Shuffle and Graph Attention Tracker(SGAT).**

score map is calculated as follows:

$$f(z, x) = \varphi(z) * \varphi(x) + b\mathbb{I}, \tag{1}$$

where $\varphi$ represents a convolution embedding function, $*$ represents the cross-correlation calculation layer, and $b\mathbb{I}$ means a signal which takes value $b \in \mathbb{R}$ in every location.

Although the cross-correlation is a simple and efficient method, it is essentially the operation of vector inner product. This is a linear matching process without using important semantic information, which makes theses trackers degrade the localization accuracy in complex scenarios, such as occlusion and motion blur. Later works [17, 46] are devoting to alleviate the negative effects of cross-correlation. Chen *et al.* [45] propose Transformer-like structural feature fusion network to replace the cross-correlation process and achieve outstanding performance.

In order to alleviate negative effects of cross-correlation, we learn a similarity measuring via a graph attention matching and shuffle attention in Siamese network. Firstly, the basic features of the template and the search region are extracted by convolution neural network. Nextly, the basic features are divided into different groups along the channel dimension, and each group is reconstructed after the channel and spatial-wise transformation. Finally, we compute the similarity between the reconstructed features of the target template and the search region. Especially, the graph attention match method takes full advantages of the structured and part-level information.

Finally, the proposed SGAT algorithm locates the target position of the current frame by classification and regression prediction head. Among them, the *cross-entropy* loss function and *IoU* loss function are used to update the model parameters by back-propagation in the classification and regression stages, respectively.

## 3.2 Shuffle attention module

Due to attention mechanisms improving CNN's ability in representing a target, they have attracted extensive attention and have been successfully applied to tracking tasks. However, existing attention mechanisms often do not fully exploit the feature dependencies of the spatial and channel dimensions and add additional overhead to the network model. Therefore, we propose an end-to-end deep model for feature extraction by combining convolutional neural

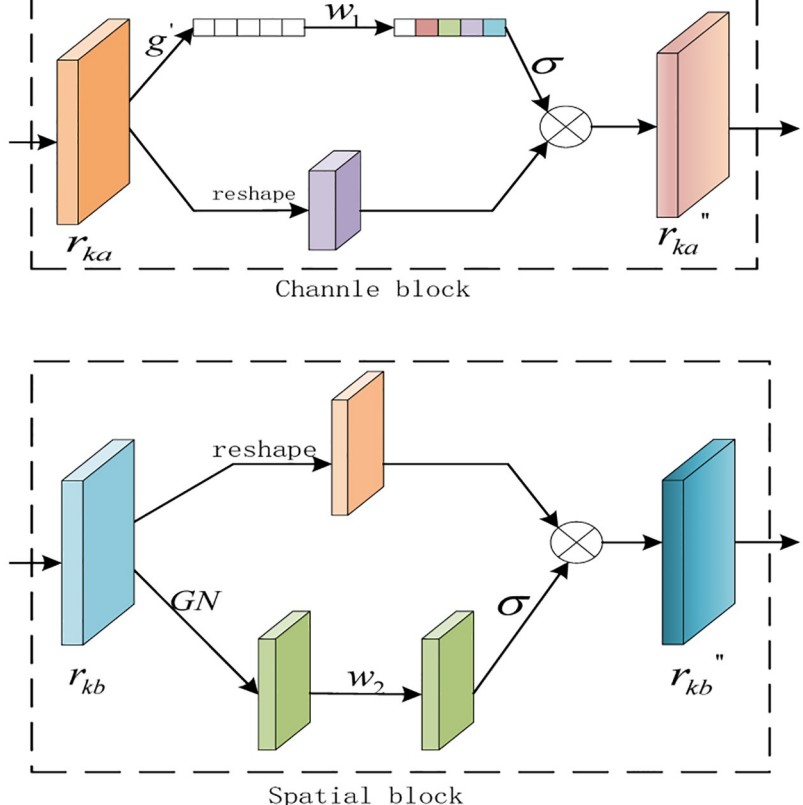

**Fig 2. The details of the channle and spatial blocks.**

networks and a shuffle attention unit. As shown in Fig 2, the designed deep model effectively exploit the correlations between the spatial and channel-wise to highlight the target region without extra overhead.

**Feature grouping**. It is assumed that there is a basis feature $r \in \mathbb{R}^{C \times H \times W}$ obtained by convolutional neural network, where $C$, $H$ and $W$ represent the channel number, height and width, respectively.

In the shuffle attention unit, a basic feature $r$ is divided into $D$ groups along the channel dimension, denoted as $r_k \in \mathbb{R}^{C/D \times H \times W}$, $k \in \{1, 2, \cdots, D\}$, where $r_k$ represents the $k$ sub-feature. In this way, the feature $r$ is divided into multiple sub-features $r_k$, and then the weight coefficient of each sub-feature is learned through off-line training. Meanwhile, each sub-feature is internally divided into two branches, $a$ and $b$, $r_k = [r_{ka}, r_{kb}]$, which uses the information correlations of spatial and channel-wise to learn the weight coefficient and reduce the redundancy of local features.

**Channel-wise transformation**. Channel transformation focuses on 'what' is important in an input image. The typical channel attention is SE block, which can effectively capture the correlation between channels. However, SE blocks usually increase the number of parameters of the model, which is not accord with the principle of lightweight design in tracking tasks. To generate channel weights efficiently, the spatial dimension of an input feature map is usually compressed, and adopt average-pooling to integrate spatial information. Based on prior information, we adopt a novel channel transformation method that resizes the channel-wise block

through global average pooling. The channel-wise block is obtained as follows:

$$r'_{kb} = \frac{1}{H \times W} \sum_{m=1}^{H} \sum_{n}^{W} r_{kb}(m, n), \tag{2}$$

where $H$ and $W$ indicates the height and width of the feature map, $r_{kb}(m, n)$ means sub-feature $r_{kb}$ at spatial location $(m, n)$.

In addition, the resized channel block is guided adaptively, and finally the final output of channel attention is obtained as follows:

$$r''_{kb} = \sigma(w_1 r'_{kb} + b) \cdot r_{kb}, \tag{3}$$

where $w_1$ and $b$ are parameters for feature scaling and shift, respectively, $\sigma$ is a *sigmoid* activation function.

**Spatial-wise transformation**. As a supplement to channel-wise transformation, spatial transformation aims to locate 'where' is an important region. To effectively carry out spatial transformation, the max-pooling and average-pooling are usually used to deal with input feature along channel dimension. In this paper, the specific implementation steps are as follows: firstly, group normalization (GN) is used to preprocess the spatial features. Then, linear transformation and activation function are combined to enhance the ability of feature representation and suppress the interference of background region. The transformed spatial features are as follows:

$$r''_{ka} = \sigma(w_2 GN(r_{ka}) + b) \cdot r_{ka}, \tag{4}$$

where $w_2$ and $b$ are parameters for feature scaling and shift, respectively, $GN$ means a group normalization.

**Reconstruction features**. After the spatial transformation, the sub-features of each group includes the spatial and channel-wise context information. Then, the sub-features are reorganized in shuffle attention unit along the channel dimension, i.e., $r = [r''_1, r''_2, \cdots, r''_k]$, and $r''_k = [r''_{ka}, r''_{kb}]$. Among them, we use *concatenate* function for sub-features reconstruction.

## 3.3 Similarity measuring

In the past, Siamese-based trackers usually use cross-correlation as similarity matching [1, 12], which is the method to match the template as a whole in the search region. However, this method is a linear matching process, which does not take advantage of nonlinear semantic information. The template block is usually represented by rectangular box as a unit, which introduces background noise into the template representation. These reasons lead to the performance bottleneck of Siamese-based tracker. To take full advantages of capture the structured and part-level information, we learn a graph attention matching based similarity measuring instead of cross-correlation. By decomposing the target template and search region features into multiple grids, and then computing the similarity of different template and search region grids, which greatly alleviate the challenging of pose variations of target. After obtaining the reconstructed features of the template and search region by a deep end-to-end model, we assume $1 \times 1 \times C$ grid of the feature map as a node. For node $i$ on the template and node $j$ in the search region, the correlation scores are:

$$e_{i,j} = f(g^i(\varphi(x)), g^j(\varphi(z))), \tag{5}$$

where $g^i, g^j$ are the reconstruct feature vetor of node $i$ and node $j$.

In order to improve the information propagate between different nodes, we adopt softmax function to normalize $e_{i,j}$ as follow:

$$\alpha_{i,j} = \frac{\exp(e_{i,j})}{\sum_{\beta \in \eta_t} \exp(e_{i,\beta})}, \tag{6}$$

where $\eta_t$ is a node set that include all template nodes.

Therefore, we obtain the discriminative feature representation as follow:

$$\hat{g}^j(\varphi(z)) = F\left(\sum_{i \in \beta_t} \alpha_{i,j} W_\nu \, \mathbf{g}^i(\varphi(x)) \parallel (W_\nu \, \mathbf{g}^j(\varphi(z)))\right), \tag{7}$$

where $W_\nu$ is a linear matrix transformation, $F(\cdot)$ is the Re$LU$ activation function.

Since the more similar the local features between the search region and the target template, the more like it is to be considered as a foreground. Therefore, we choose the inner product to measure the similarity, which is suitable to represent this relationship. The final score map is shown as:

$$f(g^i(\varphi(x)), g^j(\varphi(z))) = (w_s g^j(\varphi(z)))^T (w_z g^j(\varphi(x))), \tag{8}$$

where $w_s$ and $w_x$ are the linear transformation matrices, $g$ represents the corresponding feature vector node of the template and the search region, and $(\cdot)^T$ means the matrix transpose.

## 4 Experiments results

### 4.1 Implementation details

The proposed SGAT is implemented in Python using Pytorch on one NVIDIA Quadro P4000 GPU, Intel Xecon E5-2600 v4 CPU (2.00GHz) and 32GB RAM. Due to the limitation of insufficient computer hardware resources in the laboratory, we reduce batch size to 24. We train deep model on the training splits of GOT-10k and COCO datasets, and cut to 511*12 size through preprocessing. The sizes of search region patches and the template patch are 287*287 and 127*127, respectively. The backbone parameters are initialized with the weights that pretrained on ImageNet.

The SGAT algorithm uses GoogLeNet as the backbone network. Compared with the traditional feature extraction network, GoogLeNet can extract more richness features and use maxpooling to reduce the parameter redundancy of the upper layers.

**Evaluation metric**. We adopt the one-pass evaluation (OPE) metric of accuracy and success rate to evaluate the performance of the trackers. The precision is evaluated by the center location error (CLE) between the predicted location and the ground truth location. The precision plots are drawn in according to the frame percentages of CLE under the specified thresholds. Besides, the success rate is defined as the intersection over union (IoU) between the predicted bounding boxes and the ground truth. Meanwhile, when the IoU exceeds a certain threshold, it is considered to track the target accurately, and the success plot is drawn by the frame percentage.

### 4.2 Ablation study

To verify the effectiveness of the core components of the designed network framework, we choose OTB-100 benchmark to verify different schemes. In shuffle attention module, we divide the basis features into multiple sub-features along the channel dimensions. The shuffle unit reconstructs each sub-feature by spatial and channel-wise transformations. Finally, the

**Table 1. Ablation experiments on OTB-100 benchmark.** GM denotes graph matching, Xcorr denotes cross correlation, respectively, and SA means shuffle attention unit.

| Dataset | Backbone | Embedding Type | SA | Success rate | Precision |
|---------|----------|----------------|-----|--------------|-----------|
| OTB-100 | GoogLeNet | GM | | 0.671 | 0.855 |
| | GoogLeNet | | ✓ | **0.688** | **0.886** |
| | GoogLeNet | Xcorr | ✓ | 0.627 | 0.821 |
| | ResNet | | ✓ | 0.621 | 0.805 |

sub-features are combined by using the dependence along channel dimensions. We analyze the advantages and disadvantages of different types of backbone networks, and prove the performance gain brought by different types of backbone networks through experiments in Table 1. Meanwhile, we compare with the traditional similarity computation methods, and verify the effectiveness of graph attention matching method.

**Backbone architecture**. Reviewing the object tracking algorithms based on Siamese network, backbone networks can be mainly summarized into two kinds: 1) Shallow feature extraction networks, such as AlexNet and VggNet. The advantages of these networks are prone to converge and have small numbers of parameters. The disadvantage is that the generalization ability of the model is not enough; 2) Deep feature extraction networks, such as ResNet-152. Obviously, the deep networks can improve model ability in fitting and generalization to the data. However, the disadvantage is that there are too many parameters and the model is not lightweight enough.

In view of the above analysis, the feature extraction network we selected is GoogLeNet (Inception.V3). Compared with traditional backbone networks, such as AlexNet and ResNet, it can provide multi-scale feature fusion to increase the scale adaptability of the network, and use some tricks to reduce the feature redundancy of each layer. At the same time, we train a total of 20 epochs models to achieve convergence.

**Comparision correlation**. Existing tracking algorithms based on Siamese network use cross correlation to compute the similarity between the template patch and a search region patch, and has achieved great performance improvement. However, cross correlation is a linear matching process that neglects the structured and part-level information, which may be the bottleneck of Siamese-based trackers.

Therefore, instead of cross correlation, we use a novel graph matching to compute the similarity between the template patch and the search region patch. In this way, the similarity is calculated from part to part, which exploits the structured and part-level information. Meanwhile, we use the above shuffle attention mechanism to redistribute the weights of the features extracted from the backbone network, and highlight the target region of interest.

## 4.3 Comparision with state-of-the-art trackers

We compare the proposed SGAT with other state-of-the-art trackers, including Ocean [35], SiamFC++ [16], SiamRPN++ [14], SiamBAN [36], SiamCAR [17], ATOM [47], SPM [48], CLNet [49] on OTB-100, GOT-10k, LaSOT and UAV123. Nextly, we will analyze the tracking performance on these tracking benchmarks.

**OTB-100**. OTB-100 benchmark consists of 98 challenging video sequences, in which *Jogging* and *Skating*2 video sequences have two different initialization tracking objects, respectively. All video sequences correspond to one or more different attributes, including illumination variation (IV), occlusion (OCC), deformation (DEF), out-of-view (OV), low resolution (LR), out-of-plane rotation (OPR), in-plane rotation (IPR), fast motion (FM), background clutter (BC), motion blur (MB) and scale variation (SV), and a total of 11 attributes. As

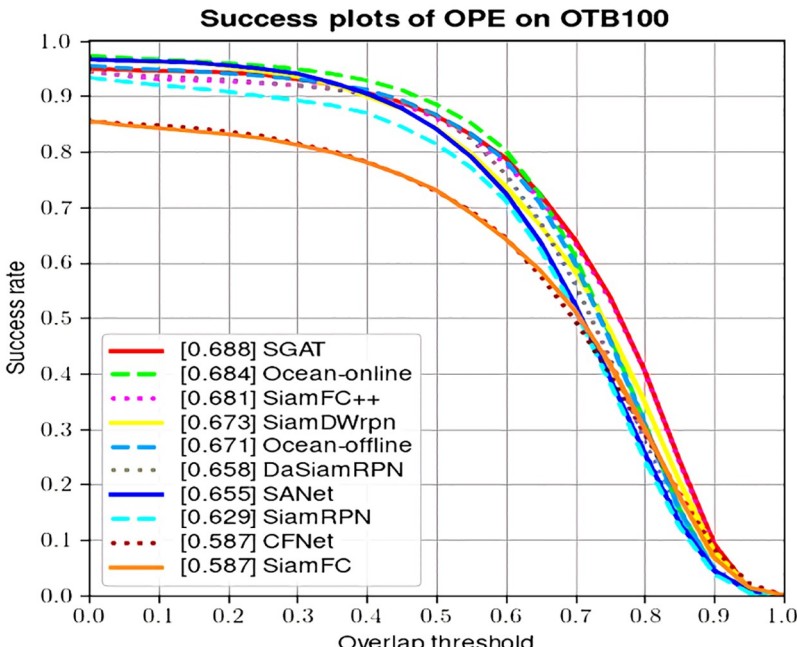

**Fig 3. Comparison with state-of-the-art trackers on OTB-100 in terms of success plots.**

shown in Fig 3, since similarity learning based on graph matching effectively exploits the structured information, the SGAT algorithm achieves the best results in the success rate and different attributes. In Fig 4, the proposed SGAT achieves an appropriate compromise between real-time speed and success rate. The results demonstrate the effectiveness of the appearance model designed by combining CNN and shuffle attention mechanism.

**GOT-10k**. GOT-10k is a recently released large-scale tracking benchmark, which includes a total of 10,000 video sequences, 563 object classes and 87 motion forms (e.g. running, swimming, skiing, crawling, cycling), and including 180 challenging video sequences in the test set. In particular, all tracking results must be evaluated in the specified server, which increases the fair contrast of the algorithm. It is worth mentioning that compared with other benchmarks, GOT-10k restricts the use of training sets for training. As shown in Table 2, we list comparisons with other the state-of-the-art trackers in terms of average overlap(AO) and success rates (SR) of thresholds 0.5 and 0.75. Success rates(SR)0.5 indicates the rate of successful tracking frames with an overlap of more than 0.5, while success rates(SR)0.75 indicates the rate of successful tracking frames with an overlap of more than 0.75. The SGAT achieves the best performance. In Fig 5, we present the AO and compared with the SOTA trackers. Benefit from the part to part similarity computation between target template and searching region, the designed algorithm achieves best performance in term of AO and success rates(SR)0.5.

**LaSOT**. LaSOT is another large-scale single object tracking dataset, which includes 1,400 video sequences and 180 test sets, with an average of more than 2,500 frames per video. LaSOT is very suitable for further evaluation of the robustness of the trackers, because long-term tracking will verify the degradation of the model and deal with some challenging factors, including occlusion, out of field of view, etc. As shown in Figs 6 and 7, the proposed SGAT achieves the best performance in terms of success rate, precision and normalized precision. In Table 3, comparing with twelve state-of-the-art trackers, the SGAT algorithm outperforms other algorithms in success rate, precision and normalized precision.

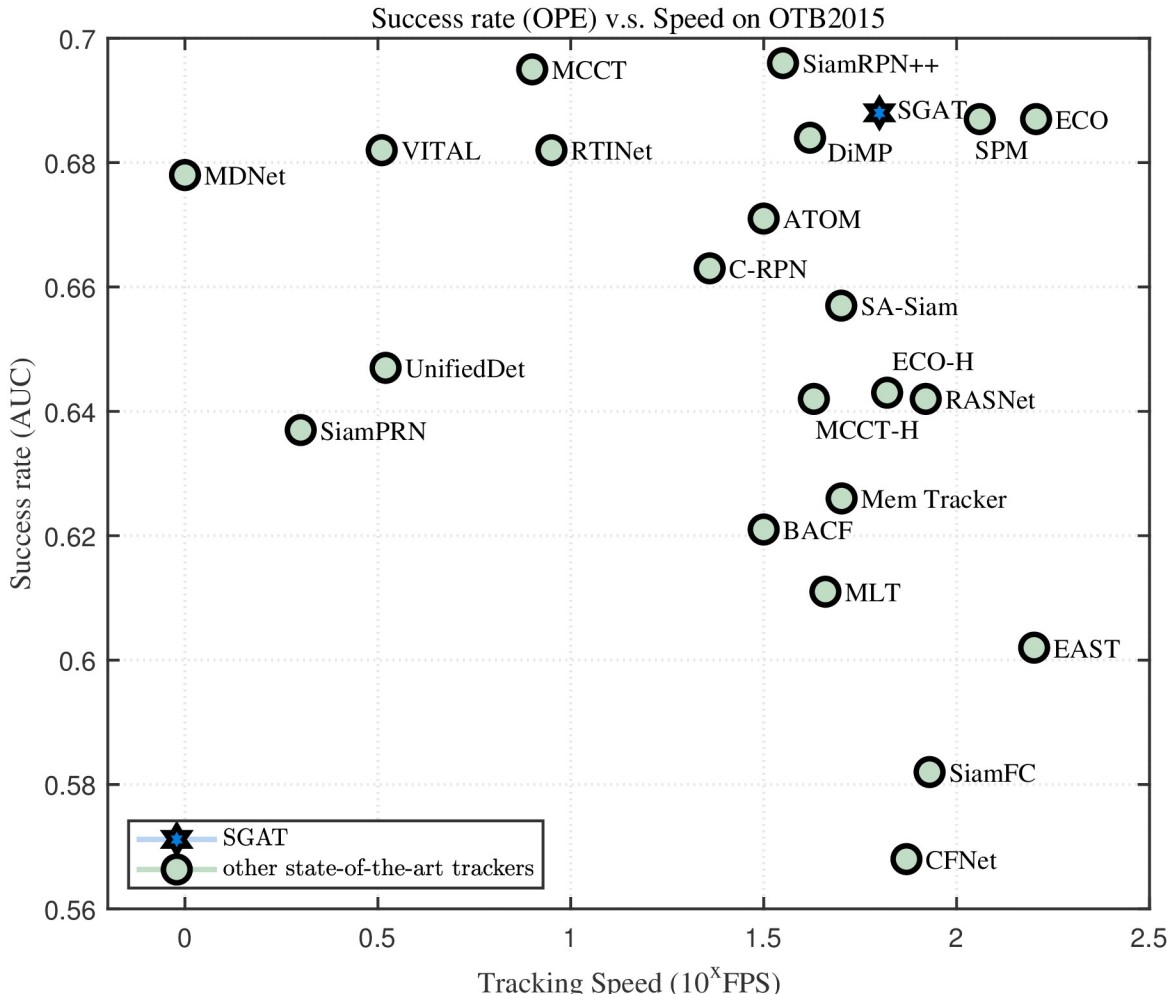

**Fig 4. Success rate vs. tracking speed on OTB-100.** Here, the x-axis represents the 10th power of the tracking speed and the y-axis represents the success rate.

**UAV123**. UAV123 is a benchmark dataset designed for UAV tracker evaluation, including 123 challenging video sequences, with an average of 935 frames per video sequence. Due to the characteristics of UAV, the main challenge factors of the test set are occlusion and small targets, and most images have low resolution attributes. In Table 4, the average overlap of the developed SGAT has reached 0.807 and the precision rate has reached 0.616, which outperform some exist mainstream tracking algorithms.

**Qualitative evaluation**. In Fig 8, we show the results compared with other trackers on four challenging sequences. At the 110th frame in the sequence *blurbody*, compared with SiamFC++ and SiamDW, due to the shuffle attention model reconstructs the basis features, the proposed SGAT can accurately regress the boundary box of the target. In the *box* and *coke* sequences, the target appears partial occlusion, accompanied by fast motion and illumination change. Compared with the tracking results of SiamFC++ and SiamDW, the SGAT algorithm can accurately regress the target bounding box. The same situation occurs in the sequence *woman*. When the target is occluded for a short time, the SGAT can locate the target again effectively. At the 488th frame, the woman is occluded by cars and trees for a short time during

**Table 2. Comparison with state-of-the-art trackers on GOT-10k benchmark.** AO, success rates(SR)0.5 and success rates(SR)0.75 represent the average overlap and the success rate at the threshold of 0.5 and 0.75.

| Trackers | AO | SR0.5 | SR0.75 |
|---|---|---|---|
| MDNet [8] | 29.9 | 30.3 | 9.9 |
| ECO [50] | 31.6 | 30.9 | 11.1 |
| CCOT [51] | 32.5 | 32.8 | 10.7 |
| SiamFC [1] | 34.8 | 35.3 | 9.8 |
| THOR [52] | 44.7 | 53.8 | 20.4 |
| SiamRPN-R18 [13] | 48.3 | 58.1 | 27.0 |
| SPM [48] | 51.3 | 59.3 | 35.9 |
| SiamRPN++ [14] | 51.7 | 61.5 | 32.9 |
| ATOM [47] | 55.6 | 63.4 | 40.2 |
| DiMP-18 [53] | 57.9 | 67.2 | 44.6 |
| SiamCAR [17] | 57.9 | 67.7 | 43.7 |
| Ocean-offline [35] | 59.2 | 69.5 | 47.3 |
| SiamFC++ [16] | 59.5 | 69.5 | 47.3 |
| **SGAT(Our)** | **59.5** | **70.1** | **46.6** |

the movement. The proposed SGAT achieves the best result on this sequence by combining the deep model and a similarity learning based on graph attention matching. Each sub-feature is reconstructed by spatial and channel transformations to highlight the target region of interest and suppress background information. In addition, a similarity learning based on graph matching can alleviate the inference of appearance variation.

**Limitations**. As shown in Fig 9, in complex tracking environment, trackers may occur tracking drift and tracking failure. In some extreme scenarios, the SGAT cannot complete the target tracking task well. For example, after the 106th frame in the sequence *soccer* and the 143rd frame in the sequence *bird*1, when there are many similar target interferences and long-

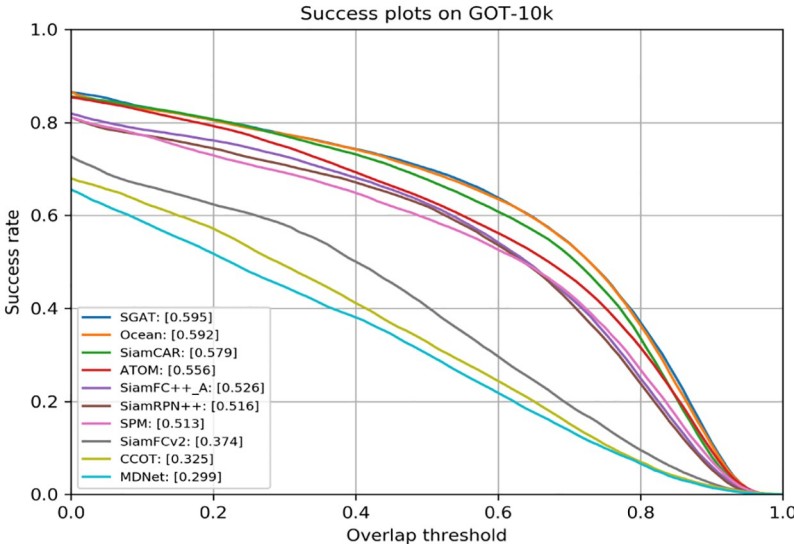

**Fig 5. Comparison with other state-of-the-art trackers on the GOT-10k benchmark, the proposed tracker SGAT achieve the best tracking performance.** A in SiamFC++_A indicates that the backbone network AlexNet.

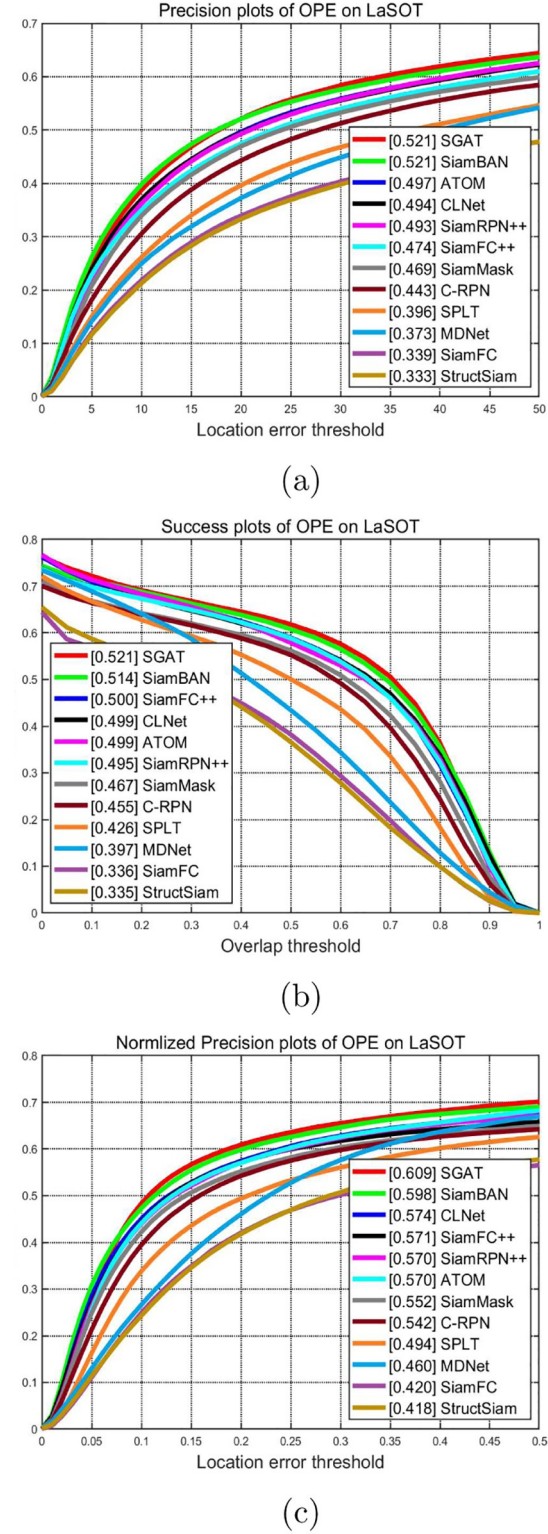

**Fig 6. Comparision with state-of-the-art trackers on LaSOT in terms of the precision rate, success rate and normalized precision plots.**

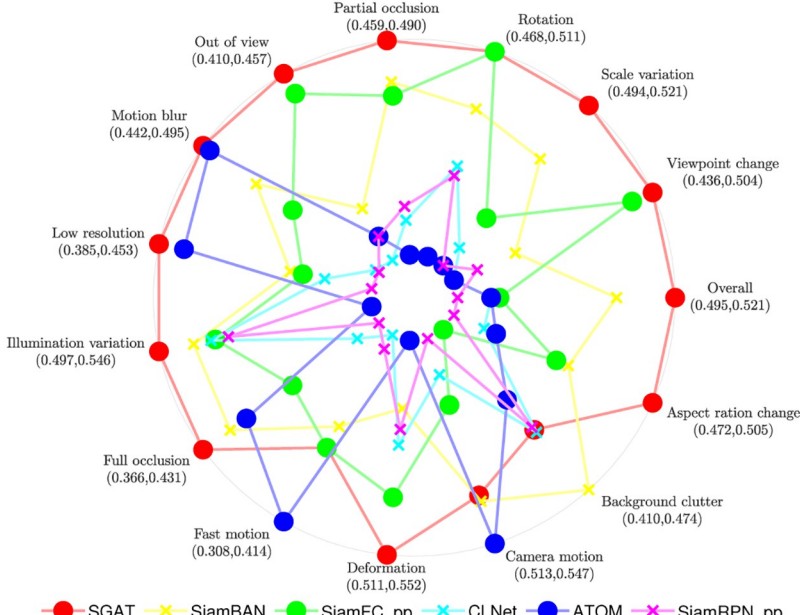

**Fig 7. Comparision with different trackers on each attribute of LaSOT.** SiamFC_pp and SiamRPN_pp represent trackers SiamFC++ and SiamRPN++, respectively.

**Table 3. A performance comparison with other competitive methods on the test split of LaSOT, where Suc., Pre. and Norm.Pre. represent the success rate, precision, normalized precision, respectively.**

| Trackers | Suc. | Pre. | Norm.Pre. |
|---|---|---|---|
| StructSiam [54] | 33.5 | 33.3 | 41.8 |
| SiamFC [1] | 33.6 | 33.9 | 42.0 |
| VITAL [55] | 39.0 | 36.0 | 48.4 |
| MDNet [8] | 39.7 | 37.3 | 46.0 |
| SPLT [56] | 42.6 | 39.6 | 49.4 |
| C-RPN [34] | 45.5 | 44.3 | 54.2 |
| SiamMask [57] | 46.7 | 46.9 | 55.2 |
| SiamFC++ [16] | 50.0 | 47.4 | 57.1 |
| SiamRPN++ [14] | 49.5 | 49.3 | 57.0 |
| CLNet [49] | 49.9 | 49.4 | 57.4 |
| ATOM [47] | 49.9 | 49.7 | 57.0 |
| SiamBAN [36] | 51.4 | 52.1 | 59.8 |
| **SGAT(Our)** | **52.1** | **52.1** | **60.9** |

**Table 4. Comparison with other competitive methods on the test split of UAV123 in terms of success rate and precision rates.**

| | Our | SiamRPN++ | DaSiamRPN | UPDT | SiamRPN | ECO | SiamFC |
|---|---|---|---|---|---|---|---|
| **Success** | **0.616** | 0.610 | 0.569 | 0.547 | 0.557 | 0.524 | 0.485 |
| **Precisionrate** | **0.807** | 0.803 | 0.781 | 0.780 | 0.768 | 0.741 | 0.693 |

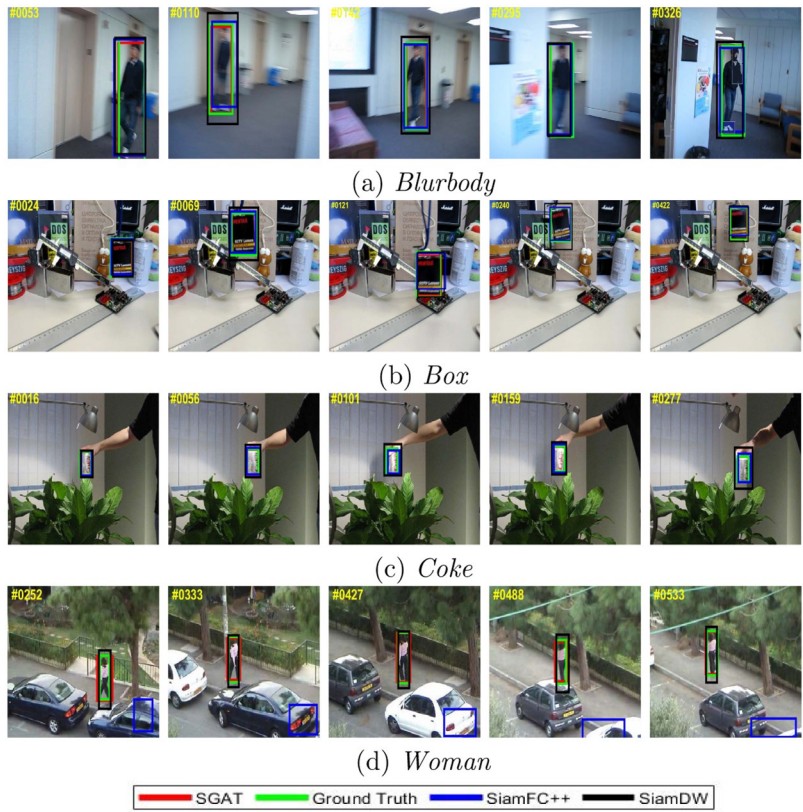

**Fig 8. Qualitative results on four challenging sequences with other state-of-the-art trackers.**

term occlusion in the scene, the SGAT will lose the target and lead to tracking failure, which is also an urgent problem faced by the existing trackers. Next, we will focus on the following two aspects: 1) modeling the target using spatial-temporal context information to ensure that the target can be located when occlusion occurs; 2) Adding a learnable memory unit to alleviate the problem that the target often disappears in long-term tracking.

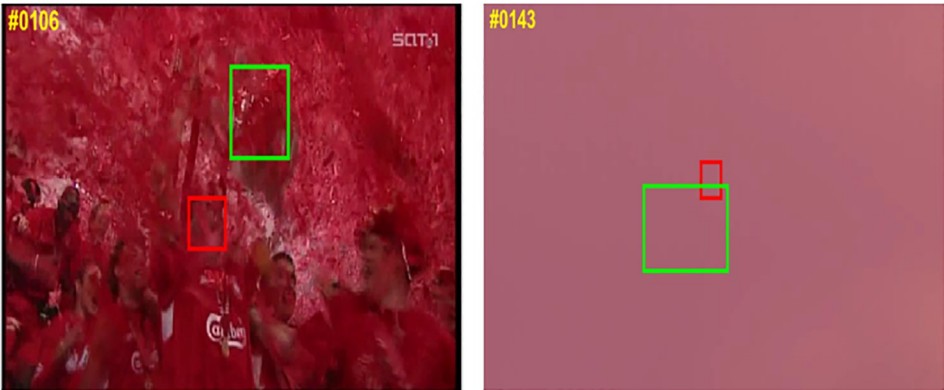

**Fig 9. Two cases of failure, in which the red box mean ground truth and the green box mean SGAT tracker.**

## 5 Conclusion

In this paper, we propose a simple and efficient visual tracking algorithm based on Siamese networks, which achieves more tracking performance in balancing between accuracy and real-time speed. In the designed deep model, the multiple sub-features are divided along the channel dimension and are processed in parallel, and take full advantage of the feature dependence of spatial and channel dimension. Sub-features are aggregated to exploit of the correlation between spatial and channel-wise information. In addition, we compute similarity between the template and a search region to obtain score map by a novel graph attention matching method, and this way effectively exploits the target structure and part-level information. Extensive experiments demonstrate that the proposed tracker achieves excellent tracking performance on multiple benchmarks and outperforming many state-of-the-art trackers.

## Acknowledgments

The authors thank the dataset providers for providing the datasets.

## Author Contributions

**Conceptualization:** Jun Wang, Yuanyun Wang.

**Data curation:** Jun Wang.

**Funding acquisition:** Yuanyun Wang, Chengzhi Deng.

**Methodology:** Yuanyun Wang, Chengzhi Deng.

**Software:** Chengzhi Deng.

**Validation:** Wenshuang Zhang.

**Visualization:** Limin Zhang.

**Writing – original draft:** Limin Zhang.

**Writing – review & editing:** Wenshuang Zhang.

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
