## [Decision Letter · Decision Letter 0]

14 Sep 2022

PONE-D-22-21523SGAT: Shuffle and Graph Attention based Siamese Networks for Visual TrackingPLOS ONE

Dear Dr. Wang,

Thank you for submitting your manuscript to PLOS ONE. After careful consideration, we feel that it has merit but does not fully meet PLOS ONE’s publication criteria as it currently stands. Therefore, we invite you to submit a revised version of the manuscript that addresses the points raised during the review process.

We look forward to receiving your revised manuscript.

Kind regards,

Sathishkumar V E

Academic Editor

PLOS ONE

Journal Requirements:

Reviewers' comments:

Reviewer's Responses to Questions

**Comments to the Author**

1. Is the manuscript technically sound, and do the data support the conclusions?

Reviewer #1: Yes

Reviewer #2: Yes

2. Has the statistical analysis been performed appropriately and rigorously? 

Reviewer #1: Yes

Reviewer #2: Yes

3. Have the authors made all data underlying the findings in their manuscript fully available?

Reviewer #1: Yes

Reviewer #2: Yes

4. Is the manuscript presented in an intelligible fashion and written in standard English?

Reviewer #1: Yes

Reviewer #2: Yes

5. Review Comments to the Author

Reviewer #1: In this paper, the authors proposed a tacking algorithm for visual tracking using shuffle and graph matching attention mechanism. Correlations between the spatial and channel-wise information to highlight the target region is explored. The results are compared with benchmark datasets. The paper is well written. The interpretation and description of the experimental results are also explained clearly. However, this manuscript have some weak points, it should be further improved before consider for publication. Some of my observations are

1. Abstract is very general. It has to be elaborated with characteristics of the results obtained.

2. No need to give specification for cl and reg in figure 1. It has to be explained through literature.

3. In table 1 also expansion for GM, SA are not needed. Expansions need to be given during first refereed place.

4. In table 1, what Success represents? What about accuracy?

5. Figure 3, more explanation is needed like the reason for SGAT algorithm’s best performance compared to all other methods

6. Figure 4 is not clear

7. Why the authors choose the threshold values 0.5 and 0.75. Justification is required

8. The need for evaluation based on AO is required

9. What is the need for representing success rate as SR? Uniformity in representing evaluation measures are not followed.

10. Table 3 need to be elaborated. It is mentioned thatUAV123 is used for evaluation. Why other state-of-the-art datasets are not used for evaluation?

11. In some places, success is represented as AUC. Need to maintain uniformity.

12. Justification is required for fig 9.

13. Recent references need to be included.

Reviewer #2: In this paper, the authors propose a shuffle attention based Siamese tracker. The idea makes sense and the paper is easy to follow. Extensive experimental results demonstrate that the proposed method achieves good performance. However, there are several problems and questions of the paper should be solved.

(1) In the abstract, the authors should introduce the core idea of the proposed method and point out the advantages of the method.

(2) The motivation is not clearly in the introduction. What's the problem of this paper solved?

(3) Why the shuffle attention is better than the original channel and spatial attentions in tracking ?

(4) There are several Siamese trackers and attentions should be discussed to enrich the related work, such as Learning dual-margin model for visual tracking, Learning Deep Multi-Level Similarity for Thermal Infrared Object Tracking, Hierarchical spatial-aware siamese network for thermal infrared object tracking, and Learning dual-level deep representation for thermal infrared tracking.

(5) What's the GM in Fig.1. There are missing the introduction about this.

(6) How to divide the group in the shuffle attention module? The authors should explain the reason and conduct an ablation study.

6. PLOS authors have the option to publish the peer review history of their article (what does this mean?). If published, this will include your full peer review and any attached files.

Reviewer #1: No

Reviewer #2: No

---

## [Author Response · Author response to Decision Letter 0]

3 Oct 2022

Paper No.: No.: PONE-D-22-21523

Title: SGAT: Shuffle and Graph Attention based Siamese Networks for Visual

Tracking

Authors: Jun Wang, Limin Zhang, Wenshuang Zhang, Yuanyun Wang*, Chengzhi 

Deng

Dear Editor:

We would like to thank the reviewers and you for your great efforts in helping us to 

improve the quality of the paper. After carefully considering the reviewers’ comments 

and suggestions, we have significantly revised the paper with more details and 

descriptions.

A detailed summary of the revisions and some specific comments/responses are given 

in the following.

In short, we feel that we have addressed all crucial concerns of the reviewers. 

However，if you have any questions or further requirements, please do not hesitate to 

contact us.

Best regards,

Yuanyun Wang

October, 3, 2022

A Response and Summary of the Revisions: No.: PONE-D-22-21523

Authors’ Responses:

Associate Editor

（1）Please ensure that your manuscript meets PLOS ONE's style requirements, 

including those for file naming. The PLOS ONE style templates can be found at

https://journals.plos.org/plosone/s/file?id=wjVg/PLOSOne_formatting_sample_main

_body.pdf and https://journals.plos.org/plosone/s/file?id=ba62/PLOSOne_formatting_sample_title_a

uthors_affiliations.pdf

RESPONSE: Thanks. We have carefully read the guidelines and style templates. 

Then, we have modified the manuscript throughout the paper according to the 

templates.

(2) Please note that PLOS ONE has specific guidelines on code sharing for 

submissions in which author-generated code underpins the findings in the manuscript. 

In these cases, all author-generated code must be made available without restrictions 

upon publication of the work. Please review our guidelines at

https://journals.plos.org/plosone/s/materials-and-software-sharing#loc-sharing-code a

nd ensure that your code is shared in a way that follows best practice and facilitates 

reproducibility and reuse.

RESPONSE: Thanks for pointing this out. We will provide the appropriate code in 

three months.

Reviewer #1

(1) Abstract is very general. It has to be elaborated with characteristics of the results 

obtained.

RESPONSE: Thanks for the comments. We have re-organized the abstract in the

revised manuscript with more details according to the results obtained.

...In this paper, we propose a novel tracking algorithm for feature extraction of target 

templates and search region images through convolutional neural networks and 

shuffle attention, and computes the similarity between the template and a search 

region through a graph attention matching. ...

…Extensive experiments demonstrate that the proposed tracking algorithm achieves 

excellent tracking results on multiple challenging benchmarks. Compared with other 

state-of-the-art methods, the proposed tracking algorithm achieves excellent tracking 

performance. …

(2) No need to give specification for cl and reg in figure 1. It has to be explained 

through literature.

RESPONSE: Thanks for the valuable suggestion. We removed the branches cls and 

reg in the revised manuscript as shown in Fig 1. We have explained both the 

classification and regression branches in Section 2, i.e., related works. In addition, this 

prediction head is the usual way of mainstream trackers, including SiamRPN[12] and 

SiamCAR[16], etc.

SiamRPN has the classification (cl) and regression (reg) branches. The classification 

branch distinguishes the target from the surrounding background. And the regression 

branch refines the target location.

(3) In table 1 also expansion for GM, SA are not needed. Expansions need to be given 

during first refereed place.

RESPONSE: Thanks for the comments. We give the expansions of GM and SA

during first refereed place.

In Table 1, we demonstrate the effectiveness of GM by ablation experiments. For 

more details, GM and SA are explained in Page 4 at the first refereed place, please see 

Section 3 and Section 4.2. In addition, we fixed the GM constant and validated the 

effectiveness of the SA module.

…2) the shuffle attention mechanism model (SA Unit), which reconstructs the basis 

features to focus on the target region and suppress the background interference 

through the spatial and channel-wise transformation; 3) graph attention matching 

(GM), which computes the similarity between the target template and a search region, 

and joints classification and regression branches to locate the target position in the 

current frame. …

(4) In table 1, what Success represents? What about accuracy?

RESPONSE: Thanks for pointing this out. In the evaluation metric, we explain the 

relevant definitions of success and accuracy. Among them, precision and accuracy

denote the same concept. For more details, please see the Section 4.1.

…The precision is evaluated by the center location error (CLE) between the predicted 

location and the ground truth location. The precision plots are drawn in according to 

the frame percentages of CLE under the specified thresholds. Besides, the success rate 

is defined as the intersection over union (IoU) between the predicted bounding boxes 

and the ground truth. Meanwhile, when the IoU exceeds a certain threshold, it is 

considered to track the target accurately, and the success plot is drawn by the frame 

percentage. …

(5) Figure 3, more explanation is needed like the reason for SGAT algorithm’s best 

performance compared to all other methods.

RESPONSE: Thanks for the suggestions. We have added some sentences for analysis 

of superior performance in Fig 3, in the revised manuscript. For more details, please 

see the Section 4.3.

…since similarity learning based on graph matching effectively exploits the structured 

information, the SGAT algorithm achieves the best results in the success rate and 

different attributes. …

(6) Figure 4 is not clear.

RESPONSE: Thanks for pointing this out. We have drawn the figure again. And we 

have added some corresponding descriptions for Figure 4. For more details, please see 

the Fig 4 and Section 4.3.

…Here, the x-axis represents the 10th power of the tracking speed and the y-axis 

represents the success rate. For example, when x is taken as 2, the tracking speed is 

200 frames per second. …

(7) Why the authors choose the threshold values 0.5 and 0.75. Justification is 

required.

RESPONSE: Thanks. In the paper associated with the GOT-10k dataset, the authors 

explicitly state the use of thresholds of 0.5 and 0.75, and subsequent mainstream 

trackers use this threshold for fair comparisons. The provision that all target trackers 

use the same training and testing sets provided by the dataset ensures a fair 

comparison of all trackers. GOT-10K training and testing sets are non-overlapping. 

After uploading the tracking results to the GOT-10K official website, the website 

automatically analyzes the tracking results. The assessment metrics provided include 

mean overlap rate (AO) and success rate (SR). SR0.5 indicates the rate of successful 

tracking frames with an overlap of more than 0.5, while SR0.75 indicates the rate of 

successful tracking frames with an overlap of more than 0.75. For more details, please 

see Section 4.3 in Page 9.

(8) The need for evaluation based on AO is required.

RESPONSE: Thanks for pointing this out.

We performed AO-based evaluation of the trackers on the GOT-10k dataset in Table 

2 in Section 4.3. Please see the corresponding explain and analysis in Section 4.3

AO represents the average overlap between all estimated bounding boxes and 

ground-truth boxes. By using AO as an evaluation index, we can further evaluate the 

tracking performance of our tracker. Meantime, we used success rate for evaluation 

on OTB2015, LaSOT and UAV123 datasets, which is IoU-based method.

(9) What is the need for representing success rate as SR? Uniformity in representing 

evaluation measures are not followed.

RESPONSE: Thanks for the suggestions. We have revised all SR to success rate 

throughout the revised manuscript.

(10) Table 3 need to be elaborated. It is mentioned that UAV123 is used for 

evaluation. Why other state-of-the-art datasets are not used for evaluation?

RESPONSE: Thanks for pointing this out.

In UAV test set, the main challenge factors are occlusion and small targets, and most 

images have low resolution attributes. To the best of our knowledge, the 

state-of-the-art trackers are usually compared on the UAV dataset. Additionally, we

also evaluate our tracker in OTB-100, GOT-10k and LaSOT. Extensive experimental 

results demonstrate that the proposed tracker has excellent performance on multiple 

benchmarks including OTB-100, GOT-10k, UAV123 and LaSOT, and outperforms 

many SOTA trackers. For more details, please see the Section 4.3.

(11) In some places, success is represented as AUC. Need to maintain uniformity.

RESPONSE: Thanks for suggestions. We revised the manuscript to maintain a 

uniform as success rate.

(12) Justification is required for fig 9.

RESPONSE: Thanks for suggestions.

Figure 9 shows the inability of our tracker to perform accurate localization in some 

complex scenarios, and we discuss it in more detail in Limitations.

... As shown in Fig 9, in complex tracking environment, trackers may occur tracking 

drift and tracking failure. In some extreme scenarios, the SGAT cannot complete the 

target tracking task well. For example, after the 106th frame in the sequence soccer

and the 143rd frame in the sequence bird1, when there are many similar target 

interferences and long-term occlusion in the scene, the SGAT will lose the target and 

lead to tracking failure, which is also an urgent problem faced by the existing trackers. 

Next, we will focus on the following two aspects: 1) modeling the target using 

spatial-temporal context information to ensure that the target can be located when 

occlusion occurs; 2) Adding a learnable memory unit to alleviate the problem that the 

target often disappears in long-term tracking. …

(13) Recent references need to be included.

RESPONSE: Thanks for comments. We added and discussed some key references in 

the revised manuscript, for example, Ref. [25], [26], [55], [56].

…In recent years, trackers based on Siamese network attract incremental attention for 

their leading performance [25-28]. …

…Fan et al. [55] propose a dual-margin model for accuracy and robust visual tracking, 

which formulated the target state prediction problem as a dual-margin model 

including an intra-object margin and an inter-object margin. Li et al. [56] propose a 

thermal infrared tracker based on a hierarchical spatially-aware twin network that 

regards the infrared tracking problem as a similarity verification task. …

[25] Hui, Le, et al. 3D Siamese Transformer Network for Single Object Tracking on 

Point Clouds. arXiv preprint arXiv:2207.11995. 2022.

[26] Tang F, Ling Q. Ranking-Based Siamese Visual Tracking. IEEE Conference 

on Computer Vision and Pattern Recognition. 2022: 8741-8750.

[55] Fan N, Li X, Zhou Z, Liu Q, He Z. Learning dual-margin model for visual

tracking. Neural Networks. 2021: 344-354.

[56] Li X, Liu Q, Fan N, He Z, Wang H. Hierarchical Spatial-aware Siamese Network

for Thermal Infrared Object Tracking. Knowledge-Based Systems. 2019: 71-81.

Reviewer #2

(1) In the abstract, the authors should introduce the core idea of the proposed method 

and point out the advantages of the method.

RESPONSE: Thanks for the comments.

We added more details in the abstract, including the core ideas of the paper and the 

advantages of the proposed approach.

...In this paper, we propose a novel tracking algorithm for feature extraction of target 

templates and search region images. Based on convolutional neural networks and 

shuffle attention, the tracking algorithm computes the similarity between the template 

and a search region through a graph attention matching. The proposed tracking 

algorithm exploits the correlations between the spatial and channel-wise information 

to highlight the target region. Moreover, the graph matching can greatly alleviate the 

influences of appearance variations such as partial occlusions. …

(2) The motivation is not clearly in the introduction. What's the problem of this paper 

solved?

RESPONSE: Thanks for pointing this out. We further explain the motivation of 

design our tracking algorithm and describe the problem solved.

a. Most Siamese-based trackers use the features of the last convolution layer or 

cascaded multi-layers as the target representations of the template and the search 

region, which do not effectively use the structured and part-level information. To 

address this problem, we propose to combine the advantages of CNN and shuffle 

attention for feature representation of target templates and search region images.

b. Both the cross-correlation and depth cross-correlation take the template features as 

a whole for linear matching on the search regions, so that the adjacent sliding 

windows produce a similar response. To solve this problem, we introduced graph 

matching approach for similarity learning to mine more structured information.

Our tracking algorithm effectively uses the structured and part-level information and 

exploits structured and part-level information, which greatly alleviate the influences 

of appearance variations such as fast motion and partial occlusions.

For more details, please see the Section 1.

(3) Why the shuffle attention is better than the original channel and spatial attentions 

in tracking?

RESPONSE: Thanks.

The original spatial and channel attention does not take full advantage of the 

correlational attention between space and channel, making it less efficient, e.g., 

CBAM. The shuffle attention by dividing into different blocks along the channel is a 

lighter and more efficient way of integrating spatial and channel attention.

(4) There are several Siamese trackers and attentions should be discussed to enrich the 

related work, such as Learning dual-margin model for visual tracking, Learning Deep 

Multi-Level Similarity for Thermal Infrared Object Tracking, Hierarchical 

spatial-aware siamese network for thermal infrared object tracking, and learning 

dual-level deep representation for thermal infrared tracking.

RESPONSE: Thanks. We have updated the literature and further enriched the related 

work as suggested.

…In recent years, trackers based on Siamese network attract incremental attention for 

their leading performance [25-28]. …

…Fan et al. [55] propose a dual-margin model for accuracy and robust visual tracking, 

which formulated the target state prediction problem as a dual-margin model 

including an intra-object margin and an inter-object margin. Li et al. [56] propose a 

thermal infrared tracker based on a hierarchical spatially-aware twin network that 

regards the infrared tracking problem as a similarity verification task. …

[27] Fan N, Li X, Zhou Z, et al. Learning dual-margin model for visual tracking.

Neural Networks. 2021: 344-354.

[28] Liu Q, Li X, He Z, et al. Learning deep multi-level similarity for thermal infrared 

object tracking. IEEE Transactions on Multimedia. 2020: 2114-2126.

[55] Fan N, Li X, Zhou Z, Liu Q, He Z. Learning dual-margin model for visual

tracking. Neural Networks. 2021: 344-354.

[56] Li X, Liu Q, Fan N, He Z, Wang H. Hierarchical Spatial-aware Siamese Network

for Thermal Infrared Object Tracking. Knowledge-Based Systems. 2019: 71-81.

(5) What's the GM in Fig 1. There are missing the introduction about this.

RESPONSE: Thanks for pointing this out. We describe the details of the GM in 

section 3.3.

…we learn a graph attention matching (GM) based similarity measuring instead of 

cross-correlation. By decomposing the target template and search region features into 

multiple grids, and then computing the similarity of different template and search 

region grids, which greatly alleviate the challenging of pose variations of target. …

... we assume 1 × 1 × C grid of the feature map as a node. For node i on the template 

and node j in the search region, the correlation scores are. …

(6) How to divide the group in the shuffle attention module? The authors should 

explain the reason and conduct an ablation study.

RESPONSE: Thanks for the comments. We have added some sentences to descript 

how to divide the group in the shuffle attention module. 

…As shown in Figure 2, the designed deep model effectively exploit the correlations 

between the spatial and channel-wise to highlight the target region without extra 

overhead. …

…Channel transformation focuses on 'what' is important in an input image. The 

typical channel attention is SE block, which can effectively capture the correlation 

between channels. However, SE blocks usually increase the number of parameters of 

the model, which is not accord with the principle of lightweight design in tracking 

tasks. To generate channel weights efficiently, the spatial dimension of an input 

feature map is usually compressed, and adopt average-pooling to integrate spatial 

information. Based on prior information, we adopt a novel channel transformation 

method that resizes the channel-wise block through global average pooling. The 

channel-wise block is obtained as follows. …

…As a supplement to channel-wise transformation, spatial transformation aims to 

locate 'where' is an important region. To effectively carry out spatial transformation, 

the max-pooling and average-pooling are usually used to deal with input feature along 

channel dimension. In this paper, the specific implementation steps are as follows: 

firstly, group normalization (GN) is used to preprocess the spatial features. Then, 

linear transformation and activation function are combined to enhance the ability of 

feature representation and suppress the interference of background region. The 

transformed spatial features are as follows. …

…In shuffle attention module, we divide the basis features into multiple sub-features 

along the channel dimensions. The shuffle unit reconstructs each sub-feature by 

spatial and channel-wise transformations. Finally, the sub-features are combined by 

using the dependence along channel dimensions. …For more details, please see the 

Section 3.2 and 4.2.

---

## [Decision Letter · Decision Letter 1]

12 Oct 2022

PONE-D-22-21523R1SGAT: Shuffle and Graph Attention based Siamese Networks for Visual TrackingPLOS ONE

Dear Dr. Wang,

Thank you for submitting your manuscript to PLOS ONE. After careful consideration, we feel that it has merit but does not fully meet PLOS ONE’s publication criteria as it currently stands. Therefore, we invite you to submit a revised version of the manuscript that addresses the points raised during the review process.

 Please include a reference to your own previously published work, https://journals.plos.org/plosone/article?id=10.1371/journal.pone.0273690 and clarify the motivations for this manuscript, in light of this previous work.

We look forward to receiving your revised manuscript.

Kind regards,

Hanna Landenmark

Staff Editor, PLOS ONE

on behalf of 

Sathishkumar V E

Journal Requirements:

Additional Editor Comments (if provided):

Reviewers' comments:

Reviewer's Responses to Questions

**Comments to the Author**

1. If the authors have adequately addressed your comments raised in a previous round of review and you feel that this manuscript is now acceptable for publication, you may indicate that here to bypass the “Comments to the Author” section, enter your conflict of interest statement in the “Confidential to Editor” section, and submit your "Accept" recommendation.

Reviewer #1: All comments have been addressed

Reviewer #2: All comments have been addressed

2. Is the manuscript technically sound, and do the data support the conclusions?

Reviewer #1: Yes

Reviewer #2: Yes

3. Has the statistical analysis been performed appropriately and rigorously? 

Reviewer #1: Yes

Reviewer #2: Yes

4. Have the authors made all data underlying the findings in their manuscript fully available?

Reviewer #1: Yes

Reviewer #2: Yes

5. Is the manuscript presented in an intelligible fashion and written in standard English?

Reviewer #1: Yes

Reviewer #2: Yes

6. Review Comments to the Author

Reviewer #1: Authors addressed all the comments specified by the reviewer. So, the paper can be accepted at this stage

Reviewer #2: The response solves my doubts.

There are two same references [27] and [55] and there is missing a related reference 'Learning dual-level deep representation for thermal infrared tracking'.

7. PLOS authors have the option to publish the peer review history of their article (what does this mean?). If published, this will include your full peer review and any attached files.

Reviewer #1: No

Reviewer #2: No

---

## [Author Response · Author response to Decision Letter 1]

13 Oct 2022

Paper No.: No.: PONE-D-22-21523R1

Title: SGAT: Shuffle and Graph Attention based Siamese Networks for Visual 

Tracking

Authors: Jun Wang, Limin Zhang, Wenshuang Zhang, Yuanyun Wang, Chengzhi 

Deng

Dear Editor:

We would like to thank the reviewers and you for your great efforts in helping us to 

improve the quality of the paper. After carefully considering the reviewers’ comments 

and suggestions, we have revised the paper with some details and references.

A detailed summary of the revisions and some specific comments/responses are given 

in the following.

In short, we feel that we have addressed all crucial concerns of the reviewers. 

However，if you have any questions or further requirements, please do not hesitate to 

contact us.

Best regards,

Yuanyun Wang

October,13, 2022

Response and Summary of the Revisions: No.: PONE-D-22-21523R1

Associate Editor

(1) Please include a reference to your own previously published work,

https://journals.plos.org/plosone/article?id=10.1371/journal.pone.0273690 and clarify 

the motivations for this manuscript, in light of this previous work.

RESPONSE: Thanks. We included the precious work [22] in Reference.

[22] Yuanyun W, Wenshuang Z, Limin Z, Jun W. Siamese network with a depthwise 

over-parameterized convolutional layer for visual tracking. PLOS ONE. 2022;1-21.

And, we added some sentences to clarity the motivation in this manuscript and the 

difference between this and the previous work.

Different from the previous work [22], we design a novel feature extraction network 

based on GoogleNet to exploit correlations of the spatial and channel-wise 

information. Additionally, in order to alleviate the influences of appearance variations, 

we use a different similarity computing to obtain more accurate score maps. Inspired 

by above-mentioned works, in this paper, we propose a novel tracking algorithm 

based on shuffle attention mechanism and graph matching in Siamese network. The 

shuffle attention mechanism in the backbone network reconstructs the basic features 

extracted from CNN, and makes the feature representation focusing on the regions of 

interest through spatial and channel-wise transformations. Different from the 

cross-correlation based similarity learning, the part-to-part graph attention matching 

further improves the tracking robustness in complex scenes, such as occlusion.

Reviewer #1

(1) Authors addressed all the comments specified by the reviewer. So, the paper can 

be accepted at this stage.

RESPONSE: Thanks for your comment.

Reviewer #2

(1) There are two same references [27] and [55] and there is missing a related 

reference 'Learning dual-level deep representation for thermal infrared tracking'.

RESPONSE: Thanks for pointing this out. We removed the same reference, and have 

included the key reference [3] in Introduction, Page 1. Some details are as follows:

[3] Liu Q, Yuan D, Fan N, et al. Learning dual-level deep representation for thermal 

infrared tracking. IEEE Transactions on Multimedia. 2022;1-8.

Visual tracking [1-3] is a fundamental research topic in computer vision. It aims to 

estimate target states in subsequent frames by given the initial state in the first frame. 

It is widely used in various applications, such as video surveillance [4], 

human-computer interaction [5], augmented reality [6], and so on. Recently, 

Convolutional Neural Network (CNN) is successfully used in visual tracking. Deep 

trackers [7,8] achieve robust tracking performance and real-time tracking speed. 

However, due to complicated appearance variations, visual tracking is still a 

challenging task.

---

## [Decision Letter · Decision Letter 2]

19 Oct 2022

SGAT: Shuffle and Graph Attention based Siamese Networks for Visual Tracking

PONE-D-22-21523R2

Dear Dr. Wang,

We’re pleased to inform you that your manuscript has been judged scientifically suitable for publication and will be formally accepted for publication once it meets all outstanding technical requirements.

Kind regards,

Sathishkumar V E

Academic Editor

PLOS ONE

Additional Editor Comments (optional):

Reviewers' comments:

Reviewer's Responses to Questions

**Comments to the Author**

1. If the authors have adequately addressed your comments raised in a previous round of review and you feel that this manuscript is now acceptable for publication, you may indicate that here to bypass the “Comments to the Author” section, enter your conflict of interest statement in the “Confidential to Editor” section, and submit your "Accept" recommendation.

Reviewer #1: All comments have been addressed

2. Is the manuscript technically sound, and do the data support the conclusions?

Reviewer #1: Yes

3. Has the statistical analysis been performed appropriately and rigorously? 

Reviewer #1: Yes

4. Have the authors made all data underlying the findings in their manuscript fully available?

Reviewer #1: Yes

5. Is the manuscript presented in an intelligible fashion and written in standard English?

Reviewer #1: Yes

6. Review Comments to the Author

Reviewer #1: All the comments have been addressed properly by the authors. The paper can be accepted at this stage.

7. PLOS authors have the option to publish the peer review history of their article (what does this mean?). If published, this will include your full peer review and any attached files.

Reviewer #1: No

---

## [Editor Report · Acceptance letter]

25 Oct 2022

PONE-D-22-21523R2 

SGAT: Shuffle and Graph Attention based Siamese Networks for Visual Tracking 

Dear Dr. Wang:

I'm pleased to inform you that your manuscript has been deemed suitable for publication in PLOS ONE. Congratulations! Your manuscript is now with our production department. 

Kind regards, 

on behalf of

Dr. Sathishkumar V E 

Academic Editor

PLOS ONE